# CNN-Based Illumination Estimation with Semantic Information

**Ho-Hyoung Choi [1], Hyun-Soo Kang [2,\*] and Byoung-Ju Yun [3,\*]**

[1] Advanced Dental Device Development Institute, School of Dentistry, Kyungpook National University, 2177, Dalgubeol-daero, Jung-gu, Daegu 41940, Korea; chhman2000@msn.com

[2] School of Information and Communication Engineering, College of Electrical and Computer Engineering, Chungbuk National University, 1, Chungdae-ro, Seowon-gu, Cheongju-si, Chungcheongbuk-do 28644, Korea

[3] School of Electronics Engineering, IT College, Kyungpook National University, 80, Daehak-ro, Buk-gu, Daegu 41566, Korea

\* Correspondence: hskang@cbnu.ac.kr (H.-S.K.); bjisyun@ee.knu.ac.kr (B.-J.Y.); Tel.: +82-53-950-7329 (B.-J.Y.)

**Abstract:** For more than a decade, both academia and industry have focused attention on the computer vision and in particular the computational color constancy (CVCC). The CVCC is used as a fundamental preprocessing task in a wide range of computer vision applications. While our human visual system (HVS) has the innate ability to perceive constant surface colors of objects under varying illumination spectra, the computer vision is facing the color constancy challenge in nature. Accordingly, this article proposes novel convolutional neural network (CNN) architecture based on the residual neural network which consists of pre-activation, atrous or dilated convolution and batch normalization. The proposed network can automatically decide what to learn from input image data and how to pool without supervision. When receiving input image data, the proposed network crops each image into image patches prior to training. Once the network begins learning, local semantic information is automatically extracted from the image patches and fed to its novel pooling layer. As a result of the semantic pooling, a weighted map or a mask is generated. Simultaneously, the extracted information is estimated and combined to form global information during training. The use of the novel pooling layer enables the proposed network to distinguish between useful data and noisy data, and thus efficiently remove noisy data during learning and evaluating. The main contribution of the proposed network is taking CVCC to higher accuracy and efficiency by adopting the novel pooling method. The experimental results demonstrate that the proposed network outperforms its conventional counterparts in estimation accuracy.

**Keywords:** human visual system (HVS); color constancy; residual neural network; semantic information; local and global information; image dataset

## 1. Introduction

The appearance of an object's color is often influenced by surface spectral reflectance, illumination condition and relative position, which makes it very challenging for the computer vision to recognize an object in both still image and video. However, the computer vision can benefit from adopting the computational color constancy (CVCC) as a pre-processing step which enables the recorded colors of the object to stay relatively constant under different illumination conditions. Obviously, color plays a large part in the performance of computer vision applications such as human computer vision, color feature extraction, and color appearance model [1,2]. However, it is imperative to cope with undesirable effects arising from the significant impact of the illumination color on the perceived color of an object in a real-life scene. While the human visual system (HVS) has the innate

ability to recognize the actual color of an object even under different light source colors, the computer vision finds it tough and challenging to identify the actual color of an object under the influence of changing illumination conditions. In an effort to mimic the HVS, the CVCC is designed to predict the actual color of an object in a real-world scene independent of varying illuminant conditions. The CVCC algorithms roughly fall into three categories: statistics-based, physics-based, and learning-based methods.

For several decades, the statics-based method has dominated the CVCC technology and the three best-known algorithms are the gray-world [3], the shades-of-gray [4] and the max-RGB (red, green, blue) [5] or the gray-edge [6]. They have a strong empirical assumption based on the statistics of real color images. There are some other statistics-based techniques which also contributed to solving the color constancy problem of the computer vision [7–9]. The physics-based method has mainly evolved from the dichromatic reflectance model of Shafer [10,11]. This method uses accurate reflection models but is still required to take complicated additional steps such as specularity estimation [10] or image segmentation [11]. The learning-based method includes Gamut mapping methods [12–15] and a recent patch-based approach [16]. The recent patch-based approach is intended to estimate the color of a light source in the local region. In this approach, the network is given a set of ground truth regions and is designed to learn and minimize their differences from the local regions. The learning-based algorithms produce state-of-the-art results but come with several drawbacks. For instance, to implement such a network, the computer is required to have a memory capable of storing thousands of patches. In addition, they need to take complicated steps to estimate local and global light sources, such as segmentation, feature extraction, and calculation of the nearest neighbor to the training set. However, Finlayson proposes the fastest learning-based method [17]. The key idea of his method is to apply the traditional gray-world assumption to estimating the color of the light source and devise a matrix to correct resultant estimation error. The network builds the matrix through dataset learning. The network learns colors and edge moments of a given image and as a result generates the elements of the matrix. Furthermore, Bianco and his colleagues [18] achieved state-of-the-art color constancy results by introducing a new method which uses a convolutional neural network. Their network has three parts: one convolutional layer for max pooling, one fully connected layer, and three output nodes. With their network, illumination estimation and fine-tuning are conducted on an image basis, not on a patch basis, and the purpose of fine-tuning is to minimize learning loss. This approach achieves a successful outcome from experimenting with one specific dataset only, so it needs to further experiment with more datasets. In addition, Lou et al. [19] proposed a deep convolutional neural network (DCNN) that is pre-trained to classify the big ImageNet dataset with labels. The performance of the network is assessed by hand-crafted color constancy algorithms. In the DCNN, ground truth labels are used to fine-tune each single dataset.

As overviewed above, there has been a decent amount of color constancy research and a number of proposed approaches. Given the structural nature of the computer vision, some challenges remain unsolved. More recently, Gijsenij et al. [20] proposed a scene semantics-based color constancy method where natural image statistics are used to identify the most important characteristics of color images. Akbarinia et al. [21] suggested a color constancy method that intends to overlap two asymmetric Gaussian kernels of different sizes in a similar way of changing the receptive field (RF) and the kernels come in different sizes depending on the contrast of surrounding pixels. Hu and colleagues [22] introduced a color constancy method which uses AlexNet and SqueezeNet in estimating illumination. Their color constancy method outperforms conventional methods by delivering state-of-the-art results. Despite their cutting-edge performance, the methodology is still with some inherent problems such as overfitting, gradient degradation, and vanishing gradient. Hussain et al. [23] proposed a color constancy method in which a histogram-based algorithm was used to determine an appropriate number of segments and efficiently split an input image into its key color variation areas. Zhan and colleagues [24] researched convolutional neural networks (CNNs) which use cross-level architecture for color constancy.

In this light, this article proposes a new network architecture-based approach and the new architecture uses the residual neural network which consists of pre-activation, atrous or dilated

convolution and batch normalization. When receiving input image data, the proposed network crops each image into image patches before training. Once the network begins training, local semantic information is automatically extracted from the image patches and fed to put its novel pooling layer. Simultaneously, the extracted information is estimated and combined to form global information during training. While conventional patch-based CNNs handle patches sequentially and individually, the proposed network takes into account all image patches simultaneously, which makes it more efficient and simpler for the network to compare and learn patches during training. The illumination estimation with the use of the image patches is formulated in this work.

Among the CNN-based color constancy approaches, some methods estimate illumination based on local image patches like the proposed approach in this work, while others rely on full image data in its entirety. In the case of the latter, the full image data comes in the form of various chroma histograms. When the network takes the full image data in chroma histograms, the convolutional filters learn to assess and identify possible illumination color solutions in chroma plane. However, spatial information is only weakly encoded in these histograms, and thus semantic context is largely ignored. When considering semantic information at the global level, it is difficult for the network to learn and discern the significance of a semantically valuable local region. To supplement this, researchers have proposed conventional convolutional networks [18,22] designed to extract and pool local features. Especially in a study by Hu and colleagues [22], the authors proposed a pooling method to extract the local confidence region from the original image and thus to form a weighted map or a mask. By using fully connected CNNs, their color constancy method shows better performance relative to its conventional counterparts. Yet it is important to challenge the estimation accuracy of the weighted map in their approach. In the fully connected layer method, each convolutional layer gets the input of all the features combined as a result of output in an earlier layer and each convolutional layer relies on local spatial coherence with a small receptive field. On the other hand, the fully connected layers have several well-known vital problems and incur incredibly high computational cost. Motivated to solve these problems, the proposed CNN method uses the residual network to improve the estimation accuracy and reduce expensive computational cost. In addition, the proposed network employs a pooling mechanism to reduce estimation ambiguities as in previous studies [18,22].

With patch processing and semantic pooling together, the proposed network is able to distinguish between useful data and noisy data during training and evaluating. In the proposed network, semantic pooling designed to extract local semantic information from the original image is performed to form a mask and the resulting image turns out a weighted map. By enabling the network to learn the semantic information in the local region and remove noisy data, the proposed color constancy approach becomes more robust to estimation ambiguities. In addition, the proposed network features end-to-end training, direct processing of arbitrary-sized images and faster computation.

To the best of our knowledge, the proposed approach is the first study to investigate and use the residual network-based CNNs to achieve color constancy. In particular, the novelty of this approach is the use of the residual network, mainly distinct from its conventional CNN-based counterparts. The residual network allows the proposed architecture to predict scene illuminant on the local region, as opposed to many previous approaches where features are extracted from the entire image to obtain statistics and estimate the overall illuminant. In addition, the dilated convolution of the residual network is designed to handle multiscale appearance, contributing to efficiency. While there are only a few methods proposed to estimate spatially varying illuminants, the proposed approach has significance and the potential to advance CNN-based illumination estimation accuracy. The experimental results demonstrate that the proposed network stays ahead of other state-of-the-art techniques in predicting illumination and is less likely to cause large errors in estimation as the conventional methods. Moreover, the proposed scheme is further applicable to solving other computer vision problems because of its strength of aggregating local estimation to determine global estimation.

## 2. Technical Approach

In recent years, deep learning techniques have become remarkably advanced and contributed to addressing computer vision challenges. The proposed network is one of the cutting-edge deep learning techniques based on ResNet [25]. ResNet is composed of several units: pre-activation, atrous convolution, batch-normalization, and layers. ResNet performs better imageNet classification when its layers use skip connection. ResNet allows researchers and developers to design much deeper networks without gradient degradation and acquire much larger receptive fields often with highly distinct features. On receiving different input images (or values), the proposed network crops each input image into image patches which carry different semantic information automatically. Next, the network learns and applies the semantic information to its novel pooling layer where all local semantic information is estimated and combined to form global information.

### 2.1. Color Constancy Approach

In general, given an RGB image, F, the color constancy approach is designed to estimate the global illuminant color $I_g = (r, g, b)$ (or color cast), using a canonical light source color, usually perfect white $\left(\frac{1}{\sqrt{3}}, \frac{1}{\sqrt{3}}, \frac{1}{\sqrt{3}}\right)^T$, and normalize the estimated global illuminant color $\hat{I}_g = \frac{I_g}{\|I_g\|}$. The approach then replaces the estimated global illuminant color with the normalized global illuminant color. However, in real-life scenes, there exist multiple illuminants, which possibly impact on the perceived color of an object. To address this problem, conventional methods attempt to estimate a single global illuminant color. Similarly, the proposed approach is designed to estimate $f_\theta$ and get the replacement, $f_\theta(F) = \hat{I}_g$, and notably uses the convolutional neural network (CNN) to estimate $f_\theta$, which gets the replacement closer to the ground truth illuminant color. $\theta$ refers to parameters.

Let $\hat{I}_g^*$ defined as the ground truth illuminant color. During dataset learning, the CNN minimizes a loss function. The loss function represents an angular error (in degrees) between the estimated color $\hat{I}_g$ and the ground truth illuminant color $\hat{I}_g^*$, described as follows:

$$L(\hat{I}_g) = \frac{180}{\pi} cos^{-1}(\hat{I}_g \times \hat{I}_g^*) \ (1) \tag{1}$$

In the CNN, $f_\theta$ is the estimation of all the semantically informative regions, ideally avoiding any repercussion of ambiguous light. Equation (2) explains how to calculate the final global illumination estimation. Let $R = \{R_1, R_2, R_3, \cdots, R_n\}$ be the local regions in $F$, and $g(R_i); i = 1, 2, 3, \cdots, n$ be the output of the regional illuminant color estimation, $R_i$. The $f_\theta$ (F) is the normalization of the sum of the product of semantic information, $c(R_i)$, and regional illumination estimation, $g(R_i)$, and as a result delivers the final global illuminant estimation color as follows [22]:

$$f_\theta(F) = \hat{I}_g = norm\left(\sum_{i \in R} c(R_i)g(R_i)\right) \ (2) \tag{2}$$

Intuitively, supposing that $R_i; i = 1, 2, 3, \cdots, n$ are local regions that contain useful semantic information for illuminant estimation, $c(R_i)$ should be large values.

In detail, the semantic pooling in Equation (2) is described as follows:

$$\hat{I}_g = \sum_{i \in R} c_i \hat{I}_i; \ i = 1, 2, 3, \cdots, n \ (3) \tag{3}$$

where

$$c_i = \sqrt{\sum_{x \in N}(\hat{I}_i)^2}; \ if \begin{cases} c_{mean,i} < color\_thresold & c_i \\ other\ wise & 0 \end{cases} \tag{4}$$

where $x$ is the coordinate of local region in the image and $N$ is the total number of pixels in the local region. $c_{mean,i}$ refers to mean of local semantic information.

$$\hat{I}_g = \frac{I_g}{\|I_g\|_2} = \frac{1}{\|I_g\|}\sum_{i \in R} c_i \hat{I}_i; \ i = 1, 2, 3, \cdots, n \tag{5}$$

where $c_i$ and $\hat{I}_i$ refer to semantic information and local illuminant estimation function, respectively.

Using the chain rule, Equation (5) is transformed into Equation (6) below:

$$\frac{\partial L(\hat{I}_g)}{\partial \hat{I}_i} = \frac{c_i}{\|I_g\|_2} \times \frac{\partial L(\hat{I}_g)}{\partial \hat{I}_g} \tag{6}$$

In Equation (6), the estimation $\hat{I}_i$ has different magnitudes with different semantic information $c_i$. In estimating local illuminants, semantic information serves as a mask within the salience region, which helps prevent the proposed network from learning noisy data. Similarly, semantic information $c_i$ is calculated as follows:

$$\frac{\partial L(\hat{I}_g)}{\partial c_i} = \frac{1}{\|I_g\|_2} \times \frac{\partial L(\hat{I}_g)}{\partial \hat{I}_g} \times \hat{I}_i \tag{7}$$

By intuition, in global estimation which uses local illumination estimation colors, it is supposed to get the global estimation color closer to the ground truth illumination color.

Figure 1 depicts a block diagram of the proposed color constancy method. As shown below, as a result of performing the proposed DCNN architecture, feature maps are generated. The feature maps turn into the weighted maps or masks through semantic pooling where the proposed network distinguishes between useful data and noisy data. The semantic pooling is formulated in Equation (2) to Equation (7). To achieve color constancy and improve the performance, it is important to pay close attention to the estimation accuracy of the proposed DCNN architecture now that it has a significant impact on the accuracy of the weighted maps and eventually on the accuracy of the global illumination estimation. In this respect, the proposed method has adopted the proposed DCNN architecture to accurately estimate the local semantic information and the accuracy is prove in the experimental results and evaluation section. The next subsection focuses on the proposed DCNN architecture.

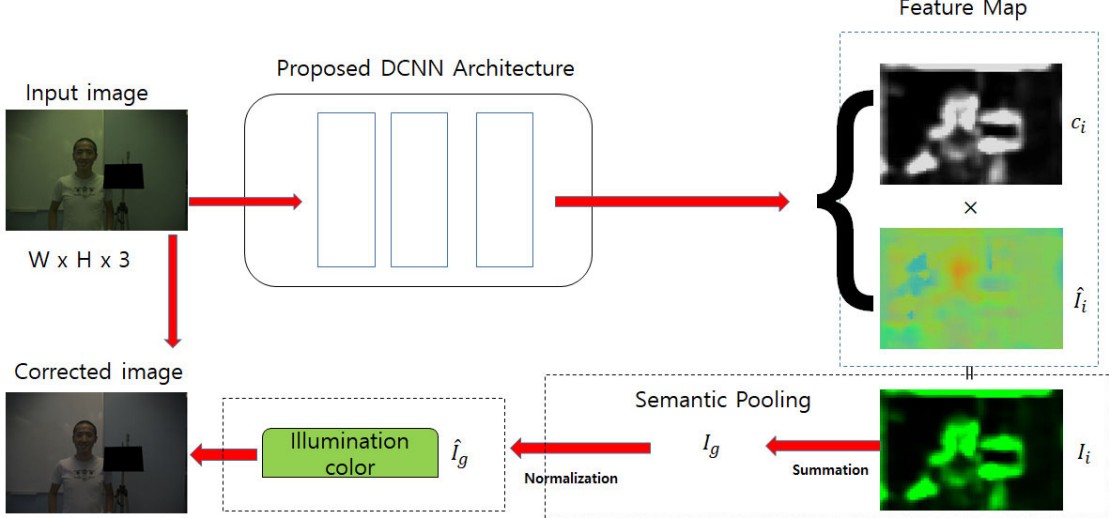

**Figure 1.** Block diagram of the proposed method.

## 2.2. The Proposed DCNN Architecture

A deep convolutional neural network (DCNN) is a major breakthrough in image classification. The DCNN naturally incorporates low, mid, and high-level image features and classifiers in an end-

to-end multi-layer form. Its depth, or the number of stacked layers, can enrich the "level" of image features. However, some deeper networks like AlexNet and VGG-16 have a degradation problem. Increasing depth of the network accelerates accuracy degradation as well as accuracy saturation. That is, an increasing number of layers cause some deeper network models to make more training errors. This is why the proposed method has adopted the well-known residual block to solve the issues with Alexnet and VGG-16 such as expensive computational cost, gradient degradation, and vanishing gradient, which are common issues in handling deep convolutional neural networks. In the proposed approach, a residual network is comprised of multiple convolutional layers. With the input of $y_{i-1}$, the output of the ith block is recursively defined as follows:

$$y_i \equiv f_i(y_{i-1}) + y_{i-1} \tag{8}$$

Let each layer take the sequential steps of convolution $f_i(x)$, batch normalization, and rectified linear unit (ReLU) as nonlinearities, and $f_i(x)$ is defined as follows:

$$f_i(x) \equiv W_i \cdot \sigma(B(W_i' \cdot \sigma(B(x)))) \tag{9}$$

where $W_i$ and $W_i'$ are weight matrices and $\cdot$ denotes convolution, $B(x)$ is batch normalization, and $\sigma(x) \equiv \max(x, 0)$. The proposed ResNet architecture shows that the resolution of feature maps drop down to a fourth of input resolution after passing through the first three layers. This allows the architecture to aggregate contexts and train faster. However, smaller feature maps constrain the architecture from learning high-resolution features which is useful and required at later stages. To support the learning of the high-resolution feature, the proposed network has an additional convolution layer with a $3 \times 3$ kernel before the first convolution layer. This enables the network to learn high-resolution features, without increasing the inference time by much. Furthermore, down-sampling principally reduces the resolution of feature maps. Although deconvolution layers are able to up-sample low-resolution feature maps, they cannot recover all the details completely. In addition, this procedure requires higher computational cost as well as intensive memory. To address such problems, the proposed method uses atrous convolution, also called dilated convolution [26]. Atrous convolution widens the kernel and simulates a larger field of perception. For a 1D input signal $x[i]$ with a filter $w[k]$ of $K$ in length, the atrous convolution is described as follows:

$$y[i] = \sum_{k=1}^{k} x[i + r \times k]w[k] \tag{10}$$

The rate $r$ refers to a stride with regard to sampling of the input signal. For instance, a rate of 2 represents a convolution on a $2 \times 2$ pooled feature map. The proposed network has changed the stride of the last convolution from 2 to 1 and set the others at $r = 2$. In this way, the smallest resolution is 16 times down-sampled, not 32 times, but still preserves the higher resolution details, as well as aggregates the usual number of contexts. Every object in a scene potentially varies in size, distance, and position. DCNN filters usually do not fit in this multiscale appearance. This has motivated researchers to investigate how DCNN [27,28] learns the multiscale feature. Their finding is that DCNN is given multi-resolution input images, which thus incur a higher computational cost. To reduce expensive computational cost and increase estimation efficiency, the proposed method gets the ResNet blocks made up of several different scale atrous convolutions with $r > 1$. In this way, the network is enabled to learn multiscale features in every block. Furthermore, the concatenation preserves all the features within the block so that the network can learn to combine features generated on different scales.

Figure 2 depicts the proposed DCNN architecture. To explain in more detail, the top half of the figure illustrates the whole process of the proposed DCNN architecture. The blue boxes are not all residual networks. There are six residual networks: two consisting of four layers and four consisting of three layers. A residual network is marked with its structure on its top right, which looks like a superscript. The bottom half of the figure gives explanatory notes and illustrates the two types of residual networks in detail. As in the explanatory notes, a convolutional layer is described in black; and the top s stands for a stride and the bottom n indicates the n by n filter kernel size, with a symbol

* to the middle left. A dilated convolution is described in red; and the top d stands for dilated convolution with a stride of 1 and the bottom n indicates the n by n filter kernel size, with a symbol * to the middle left. For instance, 1 and 1 with a symbol * in black translates as a convolutional layer with a stride of 1 and a 1 × 1 filter kernel in size. As another example, 2 and 3 with the symbol * in red mean that the rate r of dilated convolution is 2, as in Equation (10), and the filter kernel size is 3 × 3. The emphasis of using the proposed DCNN architecture is on increasing the accuracy of estimating the local semantic information, which is vital to the final performance of the network, and training the network to optimally combine the local estimates by adaptively using the corresponding g and c, as in Equation (2), for each local region, which will suppress the impact of ambiguous patches.

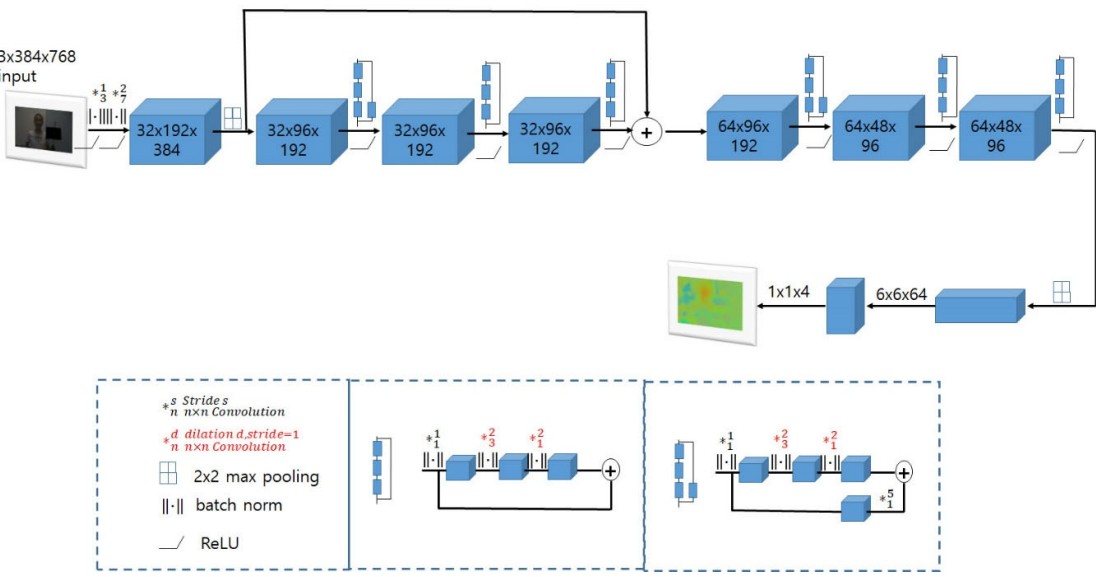

**Figure 2.** Proposed deep convolutional neural network (DCNN) architecture with pre-activation, atrous convolution, and batch-normalization.

## 3. Experimental Results and Evaluations

A feasibility study uses two benchmark standard datasets: the reprocessed [15] Color Checker Dataset [15] and the NUS 8-Camera Dataset [8]. These datasets consist of 568 and 1736 raw images, respectively. The 768 × 384 input image in Figure 2 is resized to 512 × 512 pixels and then cropped into overlapping 224 × 224 image patches. There is a trade-off between patch coverage (and accuracy) and efficiency. With more patches, the CNN performs higher coverage and accuracy, but gets lower efficiency. Through additional pooling, the proposed network combines patch-based estimates to obtain a global illuminant. The proposed network is trained in an end-to-end fashion with back-propagation. For the proposed network, Adam [29] is used to optimize parameter setting for all layers, which reduces overfitting and improves performance. The experiment with the proposed network is performed to compare total training losses at four different learning rates with 10,000 iterations (or epochs) using a server with Titan XP GPU and taking 1.5 days. Figure 3 illustrates the comparative experimental results and finds that $4 \times 10^{-4}$ is the optimized base learning rate. The symbol "1.00E-03" represents a learning rate of $1 \times 10^{-3}$. Likewise, parameters are optimized, including a dropout probability of 0.5 for the 6 × 6 × 64 convolution layer in Figure 2, a batch size of 16, a weight decay of $5 \times 10^{-5}$, and so forth.

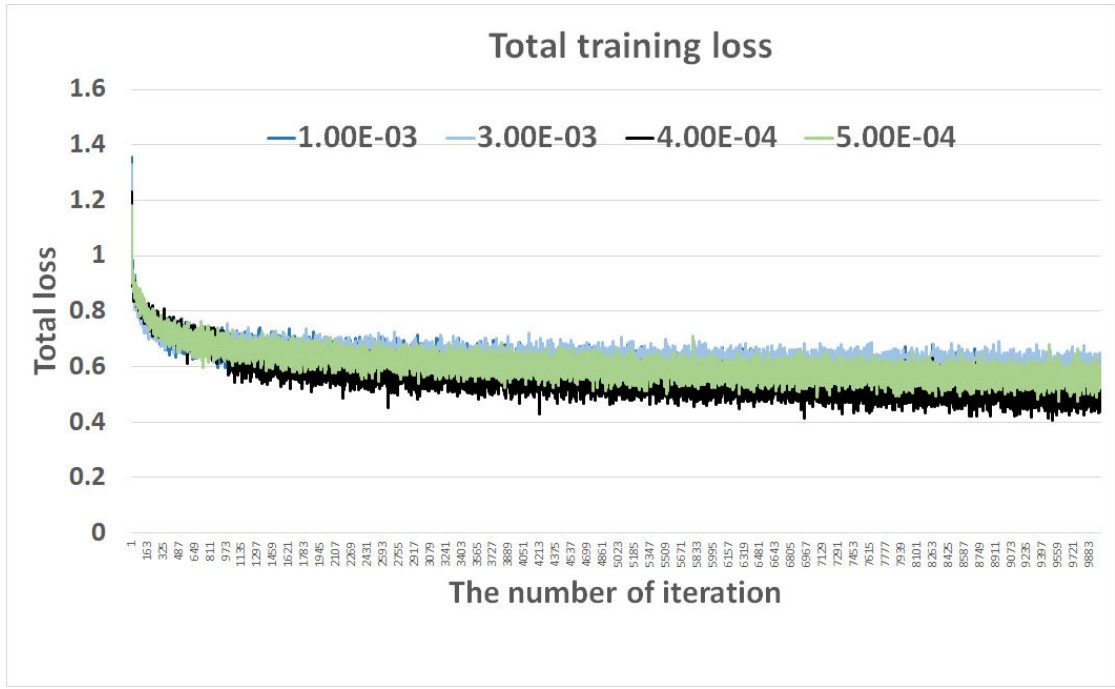

**Figure 3.** Total training loss comparison in the logarithm space of four different learning rates to optimize the base learning rate.

Figure 4 compares median angular errors with and without semantic information, recording the errors every 20 iterations (or epochs). As a result, the errors sharply drop with semantic information. From an illuminant estimation point of view, the choice of semantic information has the effect of improving computational color constancy significantly.

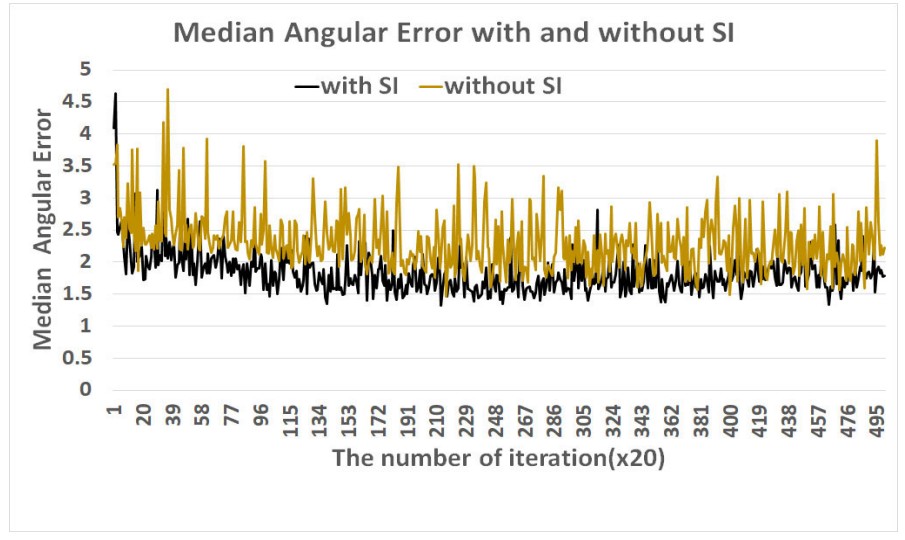

**Figure 4.** Comparison of median angular errors with and without semantic information (SI).

Figure 5 shows Shi's re-processed dataset [30] and their resulting images from implementing the proposed network in Tensorflow [31]. The proposed DCNN architecture is focused on increasing the accuracy of estimating the local semantic information to improve performance. As a result, in Figure 5e, the greenish blue illuminant of the original image is efficiently removed, and the true colors of objects are well represented without color distortion compared with the original image in Figure 5a.

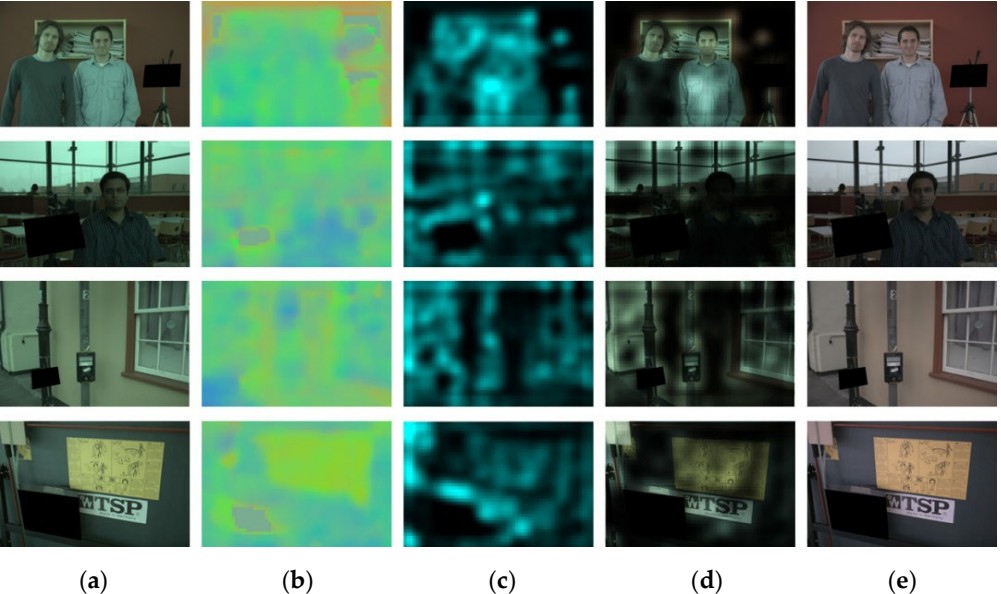

|      |      |      |      |      |
| :--: | :--: | :--: | :--: | :--: |
| (**a**) | (**b**) | (**c**) | (**d**) | (**e**) |

**Figure 5.** Shi's re-processed dataset and their resulting images: (**a**) original image, (**b**) illumination estimation map, (**c**) weighted map, (**d**) image $\times$ weighted map, and (**e**) corrected image.

The proposed network is compared with 27 different state-of-the-art methods which include both unitary and combinational methods. The 27 different methods are benchmarked from several sources. Specifically, AlexNet-FCand SqueezeNet-FC are benchmarked from [22]; and except for DS-Net, the other 22 methods are from [32–39]. DS-Net is cited from [40]. In this comparative study, the source codes of AlexNet-FC and SqueezeNet-FC are downloaded from GitHub website [41] and DS-Net is downloaded from GitHub as well [42]. The source codes of CCATI [23] and Zhan et al. [24] are implemented by MATLAB and Tensorflow, with parameters fixed as suggested by those articles.

For quantitative comparison purposes only, Table 1 compares the proposed method with previous mainstream algorithms in terms of the illuminant estimation accuracy. It illustrates several standard metrics: mean, median, trimean, mean of the best quarter (best 25%), and mean of the worst quarter (worst 25%) of angular error (Equation (1)). This comparative study uses the well-known dataset, the Gehler and Shi's dataset [15], which contains 568 images of people, places, and objects in indoor and outdoor scenes, where the Macbeth color checker chart is placed in a known location of every scene. This dataset includes both single- and multiple-illuminant natural images. The proposed network surpasses all its conventional counterparts in trimean and worst 25%.

**Table 1.** Comparative statistical metrics between the proposed network and conventional methods with Shi's re-processed dataset and the NUS-8 Camera Dataset (the lower, the better).

| Methods | Mean | Median | Trimean | Best 25% | Worst 25% |
|---|---|---|---|---|---|
| Statistics-Based Methods | | | | | |
| White patch [3] | 7.55 | 5.68 | 6.35 | 1.42 | 16.12 |
| Gray-world [32] | 6.36 | 6.28 | 6.28 | 2.33 | 10.58 |
| 1st-order grey edge [6] | 5.33 | 4.52 | 4.73 | 1.86 | 10.03 |
| 2nd-order grey edge [6] | 5.13 | 4.44 | 4.62 | 2.11 | 9.26 |
| Shades of grey [4] | 4.93 | 4.01 | 4.23 | 1.14 | 10.20 |
| General grey world [6] | 4.66 | 3.48 | 3.81 | 1.00 | 10.09 |
| Modifies white patch [7] | 3.87 | 2.84 | 3.15 | 0.92 | 8.38 |
| Bright-and-dark color PCA [32] | 3.52 | 2.14 | 2.47 | 0.50 | 8.74 |
| Local surface reflectance [34] | 3.31 | 2.80 | 2.87 | 1.14 | 6.39 |
| CCATI [23] | 2.34 | 1.60 | 1.91 | 0.49 | 5.28 |
| Learning-Based Methods | | | | | |
| SVR regression [32] | 8.08 | 6.73 | 7.19 | 3.35 | 14.89 |
| Edge-based Gamut [34] | 6.52 | 5.04 | 5.43 | 1.90 | 13.58 |
| Bayesian [34] | 4.82 | 3.46 | 3.88 | 1.26 | 10.46 |
| Pixel-based Gamut [34] | 4.20 | 2.33 | 2.91 | 0.50 | 10.72 |
| Intersection-based Gamut [34] | 4.20 | 2.39 | 2.93 | 0.51 | 10.70 |
| CART-based combination [12] | 3.90 | 2.91 | 3.21 | 1.02 | 8.27 |
| Spatio-spectral [36] | 3.59 | 2.96 | 3.10 | 0.95 | 7.61 |
| Bottom-up+ top-down [38] | 3.48 | 2.27 | 2.61 | 0.84 | 8.01 |
| ExemplarCC [38] | 2.89 | 2.27 | 2.42 | 0.82 | 5.97 |
| 19-edge corrected-moment [17] | 2.86 | 2.04 | 2.22 | 0.70 | 6.34 |
| CNN-based method [18] | 2.75 | 1.99 | 2.14 | 0.74 | 6.05 |
| Ensemble of decision tree based method [39] | 2.42 | 1.65 | 1.75 | 0.38 | 5.87 |
| Zhan et al. [24] | 2.29 | 1.90 | 2.03 | 0.57 | 4.72 |
| DS-Net [40] | 2.24 | 1.46 | 1.68 | 0.48 | 6.08 |
| SqueezeNet-FC [22] | 2.23 | 1.57 | 1.72 | 0.47 | 5.15 |
| AlexNet-FC [22] | 2.12 | 1.53 | 1.64 | 0.48 | 4.78 |
| Proposed network | **2.09** | **1.42** | **1.60** | **0.35** | **4.65** |

Figure 6 is an angular error (AE) histogram comparison between the proposed network and several best-performing conventional methods selected from Table 1: CNN-based method, CCATI, ExemplarCC, ensemble of decision (ED) tree based method, Zhan et al., DS-Net, SqueezeNet-FC, and AlexNet-FC. Joze and Drew [40] proposed an exemplar method which estimates the local source illuminant by finding the neighboring surfaces in the training data which consists of the weak color constant RGB values and the texture features. Ensemble of decision tree based method [38] is a discrete version of Gamut mapping which uses the correlation matrix, instead of the canonical Gamut for the considered illuminants, and uses the image data to calculate the probability that the illumination in the test image is caused by which of the known illuminants. Shi et al. [39] proposed a branch-level ensemble of neural networks consisting of two interacting sub-networks: a hypotheses network and a selection network. The selection network picks confident estimations from the plausible illuminant estimations generated from the hypotheses network. Shi's method produces accurate results, but the model size is huge, and its processing speed is slow. That is, when the CNNs go deeper by adding layers, fully connected layers have several well-known vital problems including incredibly expensive computational problems. To solve these problems, the proposed CNN method adopts the residual network to improve the estimation accuracy and reduce expensive computational cost. Further the pooling mechanism employed by the proposed network contributes to reducing estimation ambiguities.

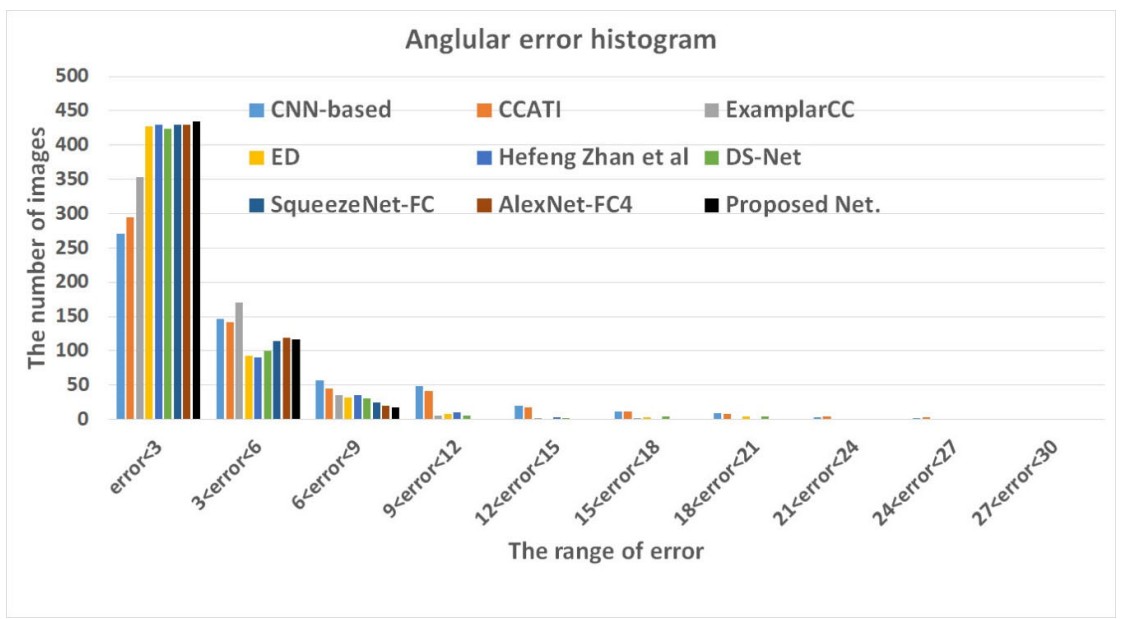

**Figure 6.** Comparative angular error (AE) histogram of convolutional neural network (CNN)-based, exemplar-based, ensemble of decision (ED) tree based methods, DS-Net, AlexNet-FC, SqueezeNet-FC, CCATI, Zhan et al., and the proposed network (proposed net.) with Shi's re-processed dataset and the NUS 8-Camera Dataset.

In estimating illuminants, the proposed network stays ahead of the state-of-the-art methods, with 76.41% of the tested images under an angular error of 3° and 97% under an angular error of 6°. Figure 7 is a comparison of the root mean square error (RMSE) results among CNN-based, exemplar-based, ensemble of decision (ED) tree based methods, Alex-FC, CCATI, Zhan et al.**,** DS-Net, SqueezeNet-FC, and the proposed network (proposed), with the input of the angular error. The proposed network records the lower RMSE relative to its conventional counterparts. Therefore, the proposed network is deemed to be robust and generates lower AE and RMSE in estimating illumination of a wide range of image scenes.

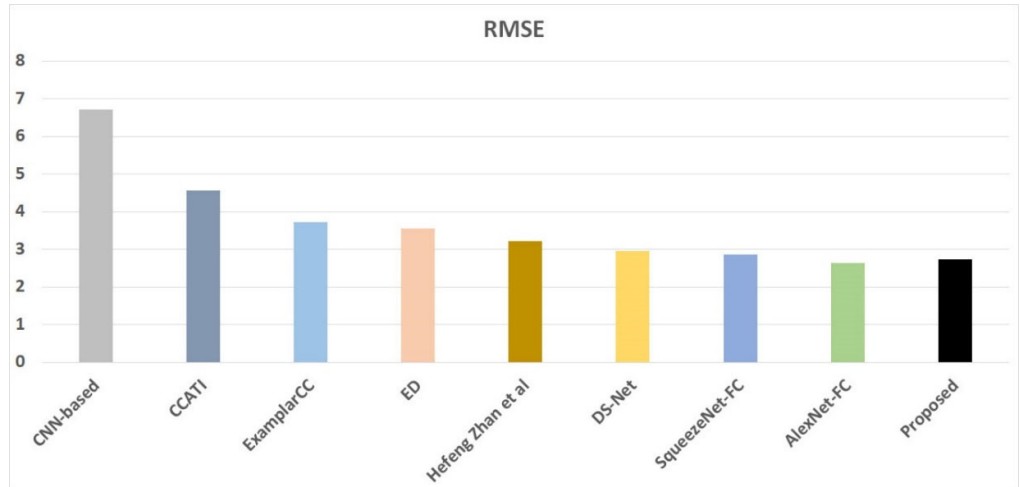

**Figure 7.** Comparison of root mean square error (RMSE) results of CNN-based, exemplar-based, ensemble of decision (ED) tree based methods, DS-Net, AlexNet-FC4, SqueezeNet-FC4, CCATI, Zhan et al., and the proposed network (proposed) with the angular error as input.

To further verify the proposed method, additional experiments were conducted using SFU-lab dataset [33] and gray-ball dataset [43]. The SFU-lab dataset contains four different subsets: objects with minimal specularities (consisting of 22 scenes, 223 images in total), objects with at least one clear dielectric specularity (9 scenes, 98 images in total), objects with metallic specularities (14 scenes, 149 images in total), and objects with fluorescent surfaces (6 scenes, 59 images in total). A commonly used subset in literature is the union of the first two subsets. Furthermore, the gray-ball dataset [43] has a total of 11,340 images of 360 × 240 pixels from a range of scenarios, which were taken under natural single- or mixed-illuminant lighting conditions and a gray-ball was placed in front of the video camera. Thus, many of the images are nearly identical scenes. Figure 8 illustrates the comparative results of median angular errors, using 321 different SFU-lab images, and Figure 9 depicts the comparative results of mean angular errors, using 500 different gray-ball dataset images. In both experimental results, the proposed network also records the lowest angular error in terms of median and mean. Therefore, the proposed method gets ahead of the conventional methods.

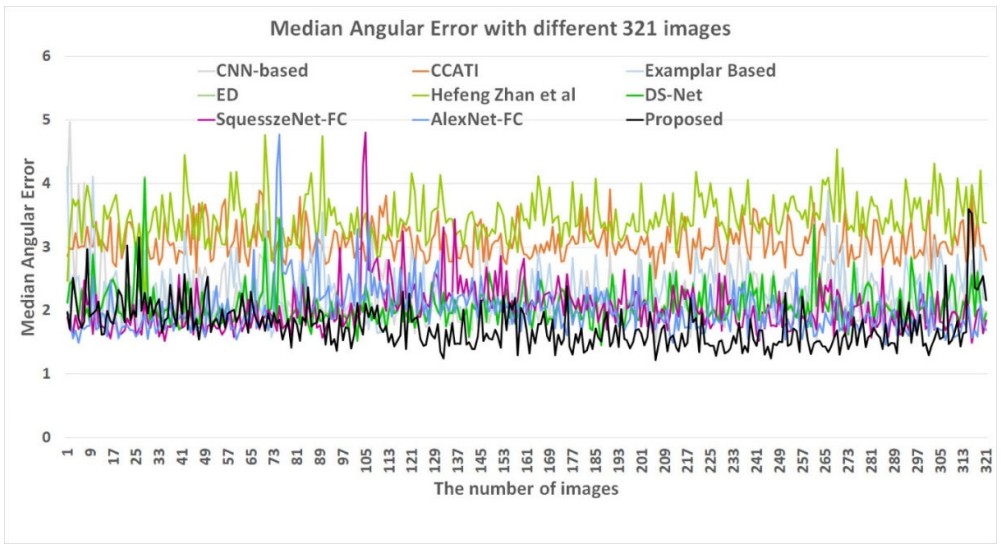

**Figure 8.** Comparison of the median angular errors of 321 different SFU-lab images.

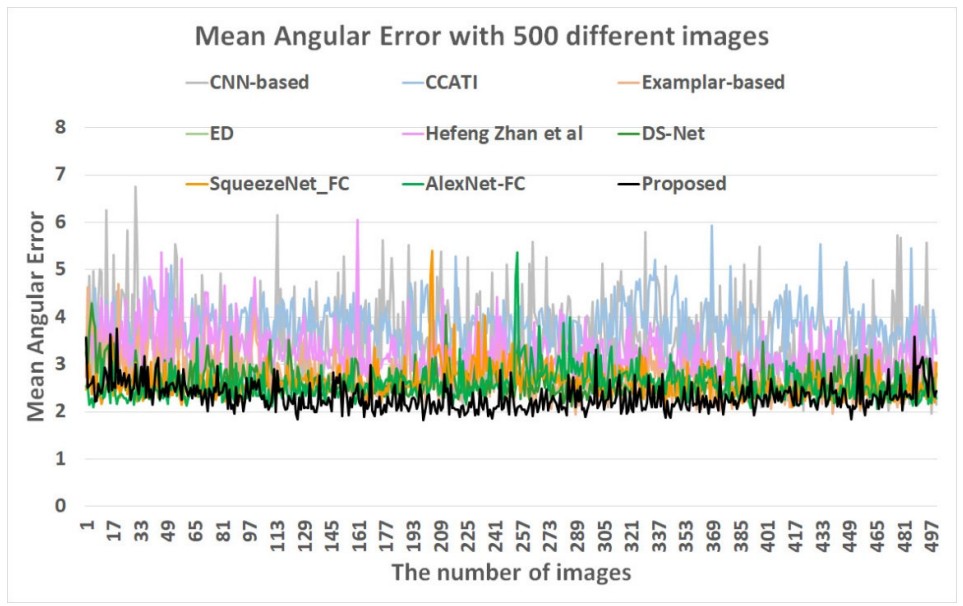

**Figure 9.** Comparison of the mean angular errors of 500 different gray-ball images.

The NUS 8-Camera Dataset [8] was additionally chosen to assess the camera invariant performance of the proposed method. The NUS 8-Camera Dataset is the most recent and well-known color constancy dataset which consists of 210 individual scenes captured by eight cameras, or a total of 1736 images. With the NUS 8-camera image dataset, 11 conventional methods and the proposed network are evaluated and compared. Table 2 displays the camera-wise performance comparison of the proposed network with the 11 conventional methods. As a result, the proposed network outperforms its 11 conventional counterparts. Accordingly, the proposed method is deemed robust regardless of camera conditions.

**Table 2.** Performance comparison between grey world (GW) [32], white patch (WP) [3], shades of grey (SoG) [4], general grey world (GGW) [6], 1st-order grey edge (GE1) [6], 2nd-order grey edge (GE2) [6], local surface reflectance statistics (LSR) [34], pixels-based Gamut (PG) [35], Bayesian framework (BF) [35], spatio-spectral statistics (SS) [37], natural image statistics (NIS) [20], and the proposed network (PN) with NUS dataset.

| | Statistics-Based | | | | | | | | Learning-Based | | | |
|---|---|---|---|---|---|---|---|---|---|---|---|---|
| Method | GW | WP | SoG | GGW | GE1 | GE2 | LSR | PG | BF | SS | NIS | PN |
| Camera | | | | | Mean Angular Error | | | | | | | |
| Canon1Ds | 5.16 | 7.99 | 3.81 | 3.16 | 3.45 | 3.47 | 3.43 | 6.13 | 3.58 | 3.21 | 4.18 | 3.18 |
| Canon600D | 3.89 | 10.96 | 3.23 | 3.24 | 3.22 | 3.21 | 3.59 | 14.51 | 3.29 | 2.67 | 3.43 | 2.35 |
| FujiXM1 | 4.16 | 10.20 | 3.56 | 3.42 | 3.13 | 3.12 | 3.31 | 8.59 | 3.98 | 2.99 | 4.05 | 3.10 |
| NikonD5200 | 4.38 | 11.64 | 3.45 | 3.26 | 3.37 | 3.47 | 3.68 | 10.14 | 3.97 | 3.15 | 4.10 | 2.35 |
| OlympEPL6 | 3.44 | 9.78 | 3.16 | 3.08 | 3.02 | 2.84 | 3.22 | 6.52 | 3.75 | 2.86 | 3.22 | 2.47 |
| LumixGX1 | 3.82 | 13.41 | 3.22 | 3.12 | 2.99 | 2.99 | 3.36 | 6.00 | 3.41 | 2.85 | 3.70 | 2.46 |
| SamNX2000 | 3.90 | 11.97 | 3.17 | 3.22 | 3.09 | 3.18 | 3.84 | 7.74 | 3.98 | 2.94 | 3.66 | 2.32 |
| SonyA57 | 4.59 | 9.91 | 3.67 | 3.20 | 3.35 | 3.36 | 3.45 | 5.27 | 3.50 | 3.06 | 3.45 | 2.33 |
| Camera | | | | | Median Angular Error | | | | | | | |
| Canon1Ds | 4.15 | 6.19 | 2.73 | 2.35 | 2.48 | 2.44 | 2.51 | 4.30 | 2.80 | 2.67 | 3.04 | 2.71 |
| Canon600D | 2.88 | 12.44 | 2.58 | 2.28 | 2.07 | 2.29 | 2.72 | 14.83 | 2.35 | 2.03 | 2.46 | 2.19 |
| FujiXM1 | 3.30 | 10.59 | 2.81 | 2.60 | 1.99 | 2.00 | 2.48 | 8.87 | 3.20 | 2.45 | 2.96 | 2.82 |
| NikonD5200 | 3.39 | 11.67 | 2.56 | 2.31 | 2.22 | 2.19 | 2.83 | 10.32 | 3.10 | 2.26 | 2.40 | 1.92 |
| OlympEPL6 | 2.58 | 9.50 | 2.42 | 2.18 | 2.11 | 2.18 | 2.49 | 4.39 | 2.81 | 2.24 | 2.17 | 2.12 |
| LumixGX1 | 3.06 | 18.00 | 2.30 | 2.23 | 2.16 | 2.04 | 2.48 | 4.74 | 2.41 | 2.22 | 2.28 | 1.42 |
| SamNX2000 | 3.00 | 12.99 | 2.33 | 2.57 | 2.23 | 2.32 | 2.90 | 7.91 | 3.00 | 2.29 | 2.77 | 1.32 |
| SonyA57 | 3.46 | 7.44 | 2.94 | 2.56 | 2.58 | 2.70 | 2.51 | 4.26 | 2.36 | 2.58 | 2.88 | 1.65 |
| Camera | | | | | Tri-mean error | | | | | | | |
| Canon1Ds | 4.46 | 6.98 | 3.06 | 2.50 | 2.74 | 2.70 | 2.81 | 4.81 | 2.97 | 2.79 | 3.30 | 2.69 |
| Canon600D | 3.07 | 11.40 | 2.63 | 2.41 | 2.36 | 2.37 | 2.95 | 14.78 | 2.40 | 2.18 | 2.72 | 2.33 |
| FujiXM1 | 3.40 | 10.25 | 2.93 | 2.72 | 2.26 | 2.27 | 2.65 | 8.64 | 3.33 | 2.55 | 3.06 | 2.88 |
| NikonD5200 | 3.59 | 11.53 | 2.74 | 2.49 | 2.52 | 2.58 | 3.03 | 10.25 | 3.36 | 2.49 | 2.77 | 1.95 |
| OlympEPL6 | 2.73 | 9.54 | 2.59 | 2.35 | 2.26 | 2.20 | 2.59 | 4.79 | 3.00 | 2.28 | 2.42 | 2.18 |
| LumixGX1 | 3.15 | 14.98 | 2.48 | 2.45 | 2.25 | 2.26 | 2.78 | 4.98 | 2.58 | 2.37 | 2.67 | 1.81 |
| SamNX2000 | 3.15 | 12.45 | 2.45 | 2.66 | 2.32 | 2.41 | 3.24 | 7.70 | 3.27 | 2.44 | 2.94 | 1.65 |
| SonyA57 | 3.81 | 8.78 | 3.03 | 2.68 | 2.76 | 2.80 | 2.70 | 4.45 | 2.57 | 2.74 | 2.95 | 1.91 |
| Camera | | | | | Mean of Best 25% | | | | | | | |
| Canon1Ds | 0.95 | 1.56 | 0.66 | 0.64 | 0.81 | 0.86 | 1.06 | 1.05 | 0.76 | 0.88 | 0.78 | 0.65 |
| Canon600D | 0.83 | 2.03 | 0.64 | 0.63 | 0.73 | 0.80 | 1.17 | 9.98 | 0.69 | 0.68 | 0.78 | 0.73 |
| FujiXM1 | 0.91 | 1.82 | 0.87 | 0.73 | 0.72 | 0.70 | 0.99 | 3.44 | 0.93 | 0.81 | 0.86 | 0.75 |
| NikonD5200 | 0.92 | 1.77 | 0.72 | 0.63 | 0.79 | 0.73 | 1.16 | 4.35 | 0.92 | 0.86 | 0.74 | 0.57 |
| OlympEPL6 | 0.85 | 1.65 | 0.76 | 0.72 | 0.65 | 0.71 | 1.15 | 1.42 | 0.91 | 0.78 | 0.76 | 0.80 |
| LumixGX1 | 0.82 | 2.25 | 0.78 | 0.70 | 0.56 | 0.61 | 0.82 | 2.06 | 0.68 | 0.82 | 0.79 | 0.65 |
| SamNX2000 | 0.81 | 2.59 | 0.78 | 0.77 | 0.71 | 0.74 | 1.26 | 2.65 | 0.93 | 0.75 | 0.75 | 0.53 |

| SonyA57 | 1.16 | 1.44 | 0.98 | 0.85 | 0.79 | 0.89 | 0.98 | 1.28 | 0.78 | 0.87 | 0.83 | 0.57 |
|---------|------|------|------|------|------|------|------|------|------|------|------|------|
| Camera | | | | | Mean of Worst 25% | | | | | | | |
| Canon1Ds | 11.00 | 16.75 | 8.52 | 7.08 | 7.69 | 7.76 | 7.30 | 14.16 | 7.95 | 6.43 | 9.51 | 6.67 |
| Canon600D | 8.53 | 18.75 | 7.06 | 7.58 | 7.48 | 7.41 | 7.40 | 18.45 | 7.93 | 5.77 | 5.76 | 5.29 |
| FujiXM1 | 9.04 | 18.26 | 7.55 | 7.62 | 7.32 | 7.23 | 7.06 | 13.4 | 8.82 | 5.99 | 9.37 | 5.64 |
| NikonD5200 | 9.69 | 21.89 | 7.69 | 7.53 | 8.42 | 8.21 | 7.57 | 15.93 | 8.18 | 6.90 | 10.01 | 4.86 |
| OlympEPL6 | 7.41 | 18.58 | 6.78 | 6.69 | 6.88 | 6.47 | 6.55 | 15.42 | 8.19 | 6.14 | 7.46 | 4.62 |
| LumixGX1 | 8.45 | 20.40 | 7.12 | 6.86 | 7.03 | 6.86 | 7.42 | 12.19 | 8.00 | 5.90 | 8.74 | 5.74 |
| SamNX2000 | 8.51 | 20.23 | 6.92 | 6.85 | 7.00 | 7.23 | 7.98 | 13.01 | 8.62 | 6.22 | 8.16 | 5.55 |
| SonyA57 | 9.85 | 21.27 | 7.75 | 6.68 | 7.18 | 7.14 | 7.32 | 11.16 | 8.02 | 6.17 | 7.18 | 5.12 |

## 4. Conclusions

A color constancy algorithm is designed to remove color casts from images and manifest the actual colors of objects, as well as preserve constant distribution of the light spectrum across the digital images, in an effort to address the challenges faced by the computer vision algorithms or methods in nature.

Accordingly, this article presents novel network architecture that uses the residual neural network composed of pre-activation, atrous or dilated convolution and batch normalization. The proposed network is intended to enable image patches to carry different semantic information automatically, upon receiving different input values. The network learns and applies semantic information to its novel pooling layer for global estimation.

As in the comparative experimental results of AE, the proposed network achieves much higher accuracy than its state-of-the-art counterparts. In the comparative AE histogram, the proposed network gets ahead of its state-of-the-art counterparts, scoring 76.41% of the number of images under an AE of 3° and 97% under an AE of 6°. In the RMSE comparison as well, the proposed network records the lowest value. Therefore, the proposed network proves to be robust and causes lower AE and RMSE in estimating illumination of a wide range of image scenes. Furthermore, through additional experiments with two more datasets of different semantic information levels: SFU-lab and gray-ball datasets, the proposed network also results in lower median and mean angular errors, respectively. In addition, the proposed network is evaluated on NUS 8-Camera Dataset to verify the camera invariant performance. As a result, the proposed method outperforms its conventional counterparts as a camera invariant color constancy model by obtaining competitive results in uniform, non-uniform, and multiple illuminant conditions. Notwithstanding, the preprocessing method and CNN structure still need to advance in estimating color casts of light sources regardless of illumination condition as well as camera sensitivity. To this end, this study will continue to advance illumination estimation accuracy.

**Author Contributions:** Conceptualization, H.-H.C., B.-J.Y., and H.-S.K.; data curation, H.-H.C.; formal analysis, H.-H.C. and B.-J.Y.; funding acquisition, B.-J.Y. and H.-S.K.; investigation, H.-H.C.; methodology, H.-H.C., B.-J.Y., and H.-S.K.; project administration, B.-J.Y. and H.-S.K.; resources, H.-H.C. and B.-J.Y.; software, H.-H.C.; supervision, B.-J.Y. and H.-S.K.; validation, H.-H.C. and B.-J.Y.; visualization, H.-S.K. and H.-H.C.; writing—original draft, H.-H.C. and B.-J.Y.; writing—review and editing, B.-J.Y. and H.-S.K. All authors have read and agreed to the published version of the manuscript.

**Acknowledgments:** This research was supported by Basic Science Research Program through the National Research Foundation (NRF) of Korea funded by the Ministry of Education (NRF-2019R1I1A3A01061844), in part by the NRF of Korea funded by the Ministry of Education (NRF-2018R1D1A1B07040457), and in part of the research projects of "Development of IoT infrastructure Technology for Smart Port" financially supported by the Ministry of Oceans and Fisheries, Korea

**Conflicts of Interest:** The authors declare no conflicts of interest.

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
