# Peer review of "CNN-Based Illumination Estimation with Semantic Information"

_applsci, doi:10.3390/app10144806_

Round 1
Reviewer 1 Report
see attached file

Author Response
Reviewer #1
Review for MDPI ‘applied sciences’ (June 13, 2020) of
CNN-based Illumination Estimation with Semantic Information
By Ho-Hyoung Choi, Hyun-Soo Kang, and Byoung-Ju Yun,
This paper proposes to use semantic information to constrain the influence of noisy image data, and to guide in the estimation of locally-differing illuminations, in correcting for illumination and recovering surface color (the ‘color constancy’ problem). The methods are interesting and appropriate, the background is clearly spelled out, and the comparisons with alternative methods are compelling. I therefore recommend publication.
P1. The main weakness in the paper is that the methods of extracting ‘semantic information’ are not spelled out, and, critically, potential errors in doing so are not discussed. I would like to see more information about this, although I do understand that most will be in the references.
<Answer> To address the reviewer’s concern, we added Eq. (4) which illustrates how the semantic information is extracted. Regarding potential errors in extracting semantic information, we had already addressed as follows.
“
To achieve color constancy and improve the performance, it is important to pay close attention to the estimation accuracy of the proposed DCNN architecture now that it has a significant impact on the accuracy of the weighted maps and eventually on the accuracy of the global illumination estimation. In this respect, the proposed method has adopted the proposed DCNN architecture to accurately estimate the local semantic information and the accuracy is proved in the experimental results and evaluation section. The next subsection focuses on the proposed DCNN architecture.
“
P2. A second major comment. The use of (r,g,b) co-ordinates for the input images is inevitable, given the data sources, but their use in the learning model did raise questions in my mind (I am a psychophysicist who studies human color constancy). These signals overlap spectrally and the primate vision system largely de-correlates them by transforming them to HSB, where H is hue (an angle in color space), S is saturation (the radial distance to white, the center) and B is brightness, the cube root of the illumination intensity. Over most intensities, B = (r+g+b)^0.33, Hue =the angle between r-g and (r+g)-k.b, where k.b=r+g at unique green, and S= chroma (the distance to white at constant B).
Physically, B varies tremendously with illumination, but color is constrained to follow a gentle curve across color space, from blue through white to orange -red, as specified by the correlated color temperature (CCT). So the task for a color constancy algorithm is to recover 3 variables, surface color (H and S) and brightness (B), despite 2-dimensional variations in illumination (intensity, and CCT).
The authors’ statement in line 164 that ‘..in real-life scenes, there exist multiple illuminants, which possibly impact on the perceived color of an object ‘ points at this issue, but really needs further explanation. In Nature, brightness varies locally, whereas in one image, color temperature does not, although CCT does vary over the day. So local light sources in an image are best treated as local variations in brightness. If lighting is natural (the Sun), then CCT and intensity (i,e., just two variables) are all that is needed to adjust. Moreover, the usual range of artificial light sources (from incandescent to a bluish-white) can also be specified by CCT. Standard colorimetry gives the transforms from (r,g,b) to (x,y, Y) to CCT. All of this (physics and visual system) elementary colorimetry is ignored in the current approach. I do understand that the paper is about how to color correct images, independent of the visual system. However I wonder why these colorimetric transformations, which match up to real-world environmental variations, are being ignored, and to what cost they are being ignored. A direct transform from the 3 classes of cones to HSB, along with adaptation, does a fair job of correcting our own vision for the illumination, without any need for learning, statistical analysis, or image segmentation. I understand that use of (r,g,b) is generic in the computational constancy field, and this must be used to permit comparisons with other methods. Nevertheless I am concerned that it is a mistake to ignore the physics (in which light sources vary in energy and in CCT; human vision is matched fairly well to this source of environmental variation. Can the various networks undo this error? Are there processing costs in doing so?
Equ. 1 refers to angle in (r, g, b), not to hue, brightness and saturation. These three have different units and if the constancy algorithm is going to minimize the effect of all three due to lighting, as is appropriate for matching to human vision, then the three must be scaled appropriately. (On a polar plot suitable for human vision, only hue is measured in angular units.) Using (r,g,b) coordinates means that brightness, hue, and saturation are not distinguished, so isolating the relevant variations in the illuminant surely becomes harder for the network.
<Answer> We do understand the reviewer’s concern over ignoring the physics. As the reviewer comments, the CCT should be considered in the color gamut mapping processing based on color metamerism. Yet the proposed method is intended to accurately estimate illumination and remove illumination, but not consider the color metamerism. However, we will consider the CCT in the further study. Thanks for your idea.
Small points:
P3. Fig.3; plotting against the logarithm of the number of iterations might make this easier to read. As it is, almost everything happens between 1 and 386, compressed to the left -hand margin.
<Answer> In response to the reviewer’s comment, we revised Fig. 3 by plotting against the logarithm of the number of iterations in order to make this easier to read.
P4. Line 326 ‘ From an illuminant estimation point of view, the choice of semantic information has the effect of improving computational color constancy significantly.’ Surely this is only true if the network correctly identifies the semantic information (e.g., a flower vase in an image, on which a local illuminant might differ due to a highlight). If the network makes errors in semantic information, the angular error should also increase. The authors should explain further how accurate their semantic recovery system is; to what extent the network was pre-trained on likely objects, and so on. Perhaps they should plot a bad example to show what happens when the semantics are incorrect.
<Answer> As in the reviewer’s comment, if the network makes errors in semantic information, the angular error should also increase. As illustrated in this article, we already addressed reviewer’s concern as follows:
“
To achieve color constancy and improve the performance, it is important to pay close attention to the estimation accuracy of the proposed DCNN architecture now that it has a significant impact on the accuracy of the weighted maps and eventually on the accuracy of the global illumination estimation. In this respect, the proposed method has adopted the proposed DCNN architecture to accurately estimate the local semantic information and the accuracy is proved in the experimental results and evaluation section. The next subsection focuses on the proposed DCNN architecture.
“
This has motivated us to propose a novel DCNN architecture.
P5. Fig. 2 legend: ‘diation’ should be ‘dilation’
<Answer> In response, we reflected the reviewer’s comment in the revised version.
P6. Fig. 4: Remark. The comparison of with and without SI is most instructive. After 150 iterations, the results stabilize. At that point, the mean angular error is 1.5 deg. with SI and 2.5 deg without SI. Thus removing SI has only a small overall effect. However, as the iterations proceed without SI, the angular error swings about, from 1.5 to 3.5 deg. To the extent that such swings are meaningful (it is impossible to tell, without specifying HSB), the learning algorithm seems defective, as one might expect to see steady progress. I do not know why the algorithm would generate such swings. However, I wonder if the learning rate can be slowed so that initial learning takes longer, but the swings are reduced, so that the no-SI results start to match up to the with SI results. If so, a large amount of computation - needed to obtain semantic information - can be by-passed.
<Answer> First of all, we wonder what makes the reviewer see the result stabilize after 150 iterations. The x-axis represents the number of iterations (x20). For example, 20 represents 400 iterations. The reviewer’s understanding seems mean angular errors 1.5 and 2.5 degrees make a small difference. However, it is not. About 1 degree difference in angular error is a huge difference. In learning, swings occur. The main point of Fig. 4 is to depict that the network with SI generates smaller angular error such as more or less 2 degrees, whereas the network without SI generates larger angular errors such as 3.5 degrees and above. Here, what is meaningful is not the swing (or oscillator) height but the angular error value. In learning, swings naturally occur. A key point is not to reduce the swing height but to lower the angular error value. To clarify another point, slowing the learning rate does not mean initial learning takes longer. The learning rate refers to the rate of convergence.
P7.Fig. 5; it might be helpful if the authors could add a column showing what happens if only the global illumination estimate is used. It looks to me as if some patches in the image will appear over-corrected, which would help make the case for the current approach. However, it is not possible to be sure, given that there are no ‘ground truth’ images in Fig. 5, i.e., (a) images of the surfaces taken under uniform white light illumination, and (b) images of the actual distribution of light sources reflected from a uniform white sheet, so the statement that the DCNN architecture ‘improves performance’ (line 339) is not obviously proven. All that can be said from Fig. 5 is that the appearance seems to be better, which relies on the viewer’s expectations as to what should be in the picture. Since Shi also published the ground truth images, I would like to see them added to Fig. 5 for comparison with the DCNN. (Adding 3 more images to Fig. 5 may require a separate figure, in which case, the last column of fig. 5 could be duplicated so as to facilitate visual comparisons across the rows.)
<Answer> Shi provides the ground truth data by using “.mat” extension from their website, but does not provide ground image. Instead, Table 1 shows the results of the angular errors from ground truth data and the estimated illumination.
P8. Table 1. The mean angular error for the proposed network is 2.30 deg (Table 1), whereas for the best statistics-based approach, CCATI, it is 2.34 deg. On the face of it, this would be a resounding success for the statistics-based approach, in that learning would seem unnecessary. However, in fig. 7, the RMSE for CCATI is 4.5 deg, compared to 2.8 deg for the proposed network, which makes it appear that the proposed network is better. I do not understand the difference. Both are being applied to the same set of images. Also, in Table 1, AlexNet-FC, DS-Net, and other learning methods also outperform the proposed network. Yet in Fig.9, the proposed network does better than AlexNet-FC and DS-Net. I don’t understand how this can be. Could there be a typo in the Table ?
<Answer> It is right that they are different. Fig. 7 is the result of the RMSE (root mean square error) of angular errors whereas Table 1 is mean, median and trimean of angular errors. Table 1 and Fig. 9 have different datasets.

Reviewer 2 Report
The authors present a CNN architecture based on residual neural network with the scope of estimating illumination based on semantic information. A path based approach is employed for illumination estimation in local regions. Furthermore this approach also allows distinguishing between useful data and noise, and thus eliminating the noise from the network training and inference. The output of the proposed algorithm is a corrected illumination image.
What is the used patch size? Do you use overlapping patches? What is the overlap?
"Each raw image is resized to 512 x 512 pixels as input patches." The input image in figure 2 is 768 x 384. This does not math with the explanations in the text. Some more clarifications are needed here to make the justification more robust.
How are the local estimations combined in the neural network, in order to obtain a global illumination estimation? Or this is not needed in the last step - obtained the corrected image (figure 1)?
The experimental section is exhaustive and covers a lot of metrics and a lot of algorithmic approaches. It would be nice to arrange the figures & tables near their explanation in the text.
The bold values in table 1 are not always the lowest values - please correct this. Table 2 is not present in the manuscript, I see only its caption.
What about real outdoor environments with different illumination and weather phenomena, such as shadows in images, fog with different densities, twilight, etc., scenes in which the distance to camera varies a lot, i.e. it is not rather constant but ranges from several meters to hundreds of meters (like traffic scenes) ? I think here the strength of the method can be really assessed.
In the paper the authors use the term "challenges faced by the computer vision". To me it sounds strange, as if computer vision is a person. I would say challenges faced by computer vision algorithms or methods.
Author Response
Reviewer #2
The authors present a CNN architecture based on residual neural network with the scope of estimating illumination based on semantic information. A path based approach is employed for illumination estimation in local regions. Furthermore this approach also allows distinguishing between useful data and noise, and thus eliminating the noise from the network training and inference. The output of the proposed algorithm is a corrected illumination image.
P1. What is the used patch size? Do you use overlapping patches? What is the overlap?
"Each raw image is resized to 512 x 512 pixels as input patches." The input image in figure 2 is 768 x 384. This does not math with the explanations in the text. Some more clarifications are needed here to make the justification more robust.
<Answer> It is our mistake to miss explaining about the overlapping patches. In this revised version, we added the following explanation:
“
The 768 x 384 input image in Fig. 2 is resized to 512 x 512 pixels and then cropped into overlapping 224x224 image patches.
“
P2.How are the local estimations combined in the neural network, in order to obtain a global illumination estimation? Or this is not needed in the last step - obtained the corrected image (figure 1)?
<Answer> We already explained about how the local estimations are combined in the neural network, which is illustrated in Eq. (2), not in Figure 1.
P3.The experimental section is exhaustive and covers a lot of metrics and a lot of algorithmic approaches. It would be nice to arrange the figures & tables near their explanation in the text.
<Answer> In response to the reviewer’s comment, we rearranged the figures near their explanation in the text. However, Tables are large in size, so it is suitable to leave as it is.
P4.The bold values in table 1 are not always the lowest values - please correct this. Table 2 is not present in the manuscript, I see only its caption.
<Answer> In response to the reviewer’s comment, we corrected the lowest values. In addition, Table 2 disappeared in the course of the associate editor converting from MS-word to PDF file. However, in MS-word format, Table 2 did appear.
P5.What about real outdoor environments with different illumination and weather phenomena, such as shadows in images, fog with different densities, twilight, etc., scenes in which the distance to camera varies a lot, i.e. it is not rather constant but ranges from several meters to hundreds of meters (like traffic scenes) ? I think here the strength of the method can be really assessed.
<Answer> The several different datasets experimented in this article such as Shi’s, SFU image lab and gray-ball datasets are widely used standard datasets in the color constancy field and already cover real outdoor environments with various illumination conditions, weather phenomena and so forth.
P6.In the paper the authors use the term "challenges faced by the computer vision". To me it sounds strange, as if computer vision is a person. I would say challenges faced by computer vision algorithms or methods.
<Answer> In response, we reflected the reviewer’s comment in the revised version.

Reviewer 3 Report
The revised manuscript has made changes according to my previous comments. I do not have more comments and concerns. I recommend this paper can be accepted this time.
Author Response
Reviewer # 3
The revised manuscript has made changes according to my previous comments. I do not have more comments and concerns. I recommend this paper can be accepted this time.
<Answer> Thanks for your recommendation for accepting this article.

This manuscript is a resubmission of an earlier submission. The following is a list of the peer review reports and author responses from that submission.
Round 1
Reviewer 1 Report
Authors have significantly rewritten the paper and addressed most of my concerns. However, there are new concerns.
- Formulas (1)-(5) look broken. Screenshot attached.
- New text in L267-268, please provide reference for sqeueezeNet-FC based color-constancy solution and Github link for implementation you used.
Thus, I think that the paper could not be published in its current state, revision recommended.

Author Response
Reviewer #1
P1. Formulas (1)-(5) look broken. Screenshot attached.
<Answer> The format trouble occurred in the course of converting the manuscript from MS word to PDF file. In response, we fixed it.
P2. New text in L267-268, please provide reference for sqeueezeNet-FC based color-constancy solution and Github link for implementation you used.
<Answer> We had already provided the reference for sqeueezeNet-FC based color-constancy solution and Github link which contains the source codes we used in implementation, but we have rewritten those sentences to avoid the reviewer’s confusion as follows:
“
Specifically, AlexNet-FC and SqueezeNet-FC are benchmarked from ref.[22]; and except for DS-Net, the other 22 methods are from ref. [34, 36]. DS-Net is cited from ref. [43]. In this comparative study, the source codes of AlexNet-FC and SqueezeNet-FC are downloaded from github website [44] and DS-Net from github website [45].
“

Reviewer 2 Report
- The abstract must be improved in terms of the evaluation part. The evaluation method in the abstract is quite vague. For example, 'outperform' in which sense and how much? Furthermore, contributions must be explicitly mentioned in the abstract.
- In the introduction part, authors need to better highlight and clearly articulate the usability and benefits of this application.
- The designed approach needs to be rewritten. The presented structure in Fig 2 is vague and somewhat generic. The structure of each block is unclear; It is not clear if they are similar to one another or not.
- The proposed method is weak and the exact added value of this work is unclear. Moreover, it is not clear what the exact 'semantic information' is in this particular application and how it is represented in the neural networks. Also, what is the semantic information role in improving the accuracy?
- The mathematical model of the input/output of the neural network is not declared.
- The format of the writing must be improved. The entire introduction is written in one single paragraph! All Equations are messed up and their format is inconsistent throughout the paper.
- Experimental results are populated with figures without proper explanation.
- Font format must be improved: Spacing between lines changes along the paper; URLs are mixed with the body of text; bold face terms appear here and there.
Author Response
Reviewer #2
P1. The abstract must be improved in terms of the evaluation part. The evaluation method in the abstract is quite vague. For example, 'outperform' in which sense and how much? Furthermore, contributions must be explicitly mentioned in the abstract.
<Answer> In response of reviewer’s comment, we have rewritten the abstract in terms of the evaluation part as follows:
“
In the proposed network, the resulting image of semantic pooling is used to develop a weighting map; the semantic information is extracted from the original image and forms a mask. The proposed network is capable of distinguishing useful data and noisy data during training and evaluation alike. In this procedure, the noise data is efficiently removed in the resulting image. The main contribution of the proposed network is to achieve higher accuracy and efficiency by using the novel pooling method. The experimental results show that the proposed network improves accuracy by 10% in predicting scene illumination compared with its counterparts. “.
P2. In the introduction part, authors need to better highlight and clearly articulate the usability and benefits of this application.
<Answer> In the introduction part, the usability and benefits of color constancy and the proposed method were already described as follows:
“
However, the computer vision can benefit from adopting the CVCC as a pre-processing step, which ensures the recorded colors of the object stay constant under different illumination conditions. Without doubt, color is an important element in performing computer vision applications such as human computer vision, color feature extraction and color appearance model [1], [2].
“
and
“
Like that, the network learns color constancy information from the local areas of the images in the dataset, and also learns to combine that information to make the final estimation result. With patch processing and semantic pooling together, the proposed network is able to distinguish between useful data and noisy data during training and evaluation alike, and has additional merits such as end-to-end training, direct processing of arbitrary-sized images and faster computation.
“
In the following sections, they are better highlighted and clearly articulated by empirical, comparative studies and experimental results.
P3. The designed approach needs to be rewritten. The presented structure in Fig 2 is vague and somewhat generic. The structure of each block is unclear; It is not clear if they are similar to one another or not.
<Answer> Fig 2 is a straightforward manifestation of how the network proceeds. The figure tells itself. This is how networks are expressed in this field. Why does the reviewer say the structure is vague and somewhat generic? It does not make sense to describe a network as vague or generic because the purpose of network is efficient color constancy performance and depending on how to compose networks their performance varies. Does the reviewer really understand the network?
In response to the reviewer’s comment P5 below, we added Eq. (7) and (8) and some explanatory sentences, which will help the reviewer understand the structure of each block. To explain in more detail, in Fig 2, the whole process of the network is presented at the top and it contains total six residual networks: two 4-layer residual networks and four 3-layer residual networks. In the whole process at the top, a residual network is marked with its structure on its top right, which looks like a superscript. At the bottom, residual networks are in detail expressed and there are two types of residual networks: 3-layer residual networks and 4-layer residual networks. Back to the top, the whole process includes two 4-layer residual networks and four 3-layer residual networks. Will this help the reviewer better understand?
P4. The proposed method is weak and the exact added value of this work is unclear. Moreover, it is not clear what the exact 'semantic information' is in this particular application and how it is represented in the neural networks. Also, what is the semantic information role in improving the accuracy?
<Answer> The proposed method aims to accurately and effectively estimate scene illumination, which helps the computer vision achieve color constancy and recognize objects. The value of this work is already and exactly presented in the introduction part, described as follows:
“
However, the computer vision can benefit from adopting the CVCC as a pre-processing step, which ensures the recorded colors of the object stay constant under different illumination conditions. Without doubt, color is an important element in performing computer vision applications such as human computer vision, color feature extraction and color appearance model [1], [2].
“
The reviewer’s words “the exact added value” can be interpreted in different meanings. If the words simply mean an improvement or addition to something that makes it worth more, the authors have already demonstrated the exact improvement of this work in comparison with 27 state-of-the-art methods in Table 1 and several Figures. Or if the reviewer’s words intend to mean an increase in value, in terms of economics, as a result of applying this work to related applications, the reviewer’s comment is not suitable and relevant to the academic paper review process, we believe. However, the authors can say that this work absolutely has value as already stated above. Any image processing experts can easily recognize the value of this work.
Next, why does the reviewer ask again what the semantic information is and how it is represented in the neural networks? They are already represented in the manuscript as follows, and the reviewer once asked the same question as of 02 Mar 2020 (date of this review) and we once answered.
“
Figure 5 shows Shi’s Re-processed dataset [29] and their resulting images from implementing the proposed network in Tensorflow [33]. (a) is the original image. (b) is the resulting image which reflects illumination estimation of the original image. (c) is the weighted map which represents semantic information in the salience region; and the semantic information is extracted from the original image and forms a mask. (d) is the resulting image which equals the product of the original image and the weighted map. (e) is the corrected image from removing illumination estimation of the original image.
”
To explain in more detail, Fig. 5(c) shows what the exact semantic information is and how semantic information is represented in the neural networks. In the proposed network, semantic information is extracted from the original image and goes through the semantic pooling. The resulting image from the semantic pooling is the weighted map and the weighted map is used to form a mask. Therefore, the weighted map represents the semantic information in the salience region. The proposed network is capable of distinguishing useful data and noisy data during training and evaluation alike. In this procedure, the noise data is efficiently removed in the resulting image. This is how the semantic information plays a role in improving the accuracy.
P5. The mathematical model of the input/output of the neural network is not declared.
<Answer> In response to the reviewer’s comment, we added the mathematical models of the input/output of the convolutional neural network, Eq. (7) and Eq.(8).
P6. The format of the writing must be improved. The entire introduction is written in one single paragraph! All Equations are messed up and their format is inconsistent throughout the paper.
<Answer> The format trouble occurred in the course of converting the manuscript from MS word to PDF file. In response, we fixed it. Also, we have divided the introduction into several paragraphs.
P7. Experimental results are populated with figures without proper explanation.
<Answer> Experimental results are demonstrated in tables and figures which tell much about the performance already with proper analytic explanation as follows.
“
Figure 6 is AE (angular error) histogram comparison between several best-performing conventional methods selected from Table 1 and the propose network. In estimating illuminants, the proposed network stays ahead of the state-of-the-art methods, having 76.41% of the tested images under an angular error of and 97% under an angular error of . Figure 7 is a comparison of RMSE results among CNN-based, Exemplar-based, Ensemble of Decision (ED) tree based-methods, Alex-FC, CCATI, Hefeng Zhan et al, DS-Net, squeezeNet-FC and the proposed network (Proposed), with the angular error as an input.
“.
P8. Font format must be improved: Spacing between lines changes along the paper; URLs are mixed with the body of text; bold face terms appear here and there.
<Answer> We improved the font format along the paper, but spacing between lines changes along the paper as a result of putting the equations or symbols. In addition, we separated URLs from the body of text and moved them to the reference in response.

Round 2
Reviewer 2 Report
The changes and the response letter have not convincingly addressed the main concerns mentioned in the first round of review.
While the manuscript contains some marginal novelty, it does not sound enough and cannot be recommended for a publication, unless clearly and explicitly highlighted and articulated (please see the comments on first round revision again). Also, the practical motivation, and the description of the methodology, are not clear and competently explained.
In view of the above, I do not recommend the paper for publication, unless the concerns are addressed fundamentally (rather than superficially).
Author Response
P1. While the manuscript contains some marginal novelty, it does not sound enough and cannot be recommended for a publication, unless clearly and explicitly highlighted and articulated (please see the comments on first round revision again). Also, the practical motivation, and the description of the methodology, are not clear and competently explained.
<Answer> In response to the reviewer’s comment, we further highlighted and articulated the usability and benefits of this application and practical motivation as follows:
“
Among the CNN-based color constancy approaches, some methods estimate illumination based on local image patches like the proposed approach in this work, while others do based on full image data in entirety. In the case of the latter, the full image data comes in the form of various chroma histograms. When the network takes the full image data in chroma histograms, the convolutional filters learn to assess and identify possible illumination color solutions chroma plane. However, spatial information is only weakly encoded in these histograms, and thus semantic context is largely ignored. When it comes to considering semantic information at the global level, it is difficult for the network to learn and discern the significance of semantically valuable local region. To supplement this, a conventional convolutional network [18] designed to extract and pool local features is proposed. Motivated by the approach of extracting and pooling local features, the proposed CNN method employs a pooling mechanism to cope with estimation ambiguities. With patch processing and semantic pooling together, the proposed network is able to distinguish between useful data and noisy data during training and evaluating. In the proposed network, semantic pooling designed to extract local semantic information from the original image is performed to form a mask and the resulting image turns out a weighted map. By enabling the network to learn the semantic information in the local region and remove noisy data, the proposed color constancy approach becomes more robust to estimation ambiguities.
”
Is the above enough for the reviewer to better understand?

Round 3
Reviewer 2 Report
Thanks for the further clarification.
I am not convinced that the paper has enough of novelty and impact to be published as a journal article.